# N-Terminal α-Helices in Domain I of *Bacillus thuringiensis* Vip3Aa Play Crucial Roles in Disruption of Liposomal Membrane

**DOI:** 10.3390/toxins16020088

**Published:** 2024-02-06

**Authors:** Ensi Shao, Hanye Huang, Jin Yuan, Yaqi Yan, Luru Ou, Xiankun Chen, Xiaohong Pan, Xiong Guan, Li Sha

**Affiliations:** 1China National Engineering Research Center of JUNCAO Technology, College of Life Science, Fujian Agriculture and Forestry University, Fuzhou 350002, China; es776@fafu.edu.cn (E.S.); yuanjin8956@126.com (J.Y.); yyq961202@163.com (Y.Y.); olr13395087102@163.com (L.O.); cxiankun@126.com (X.C.); 2State Key Laboratory of Ecological Pest Control for Fujian and Taiwan Crops & Key Laboratory of Biopesticide and Chemical Biology of Ministry of Education, Fujian Agriculture and Forestry University, Fuzhou 350002, China; hhy2460838406@163.com (H.H.); panxiaohong@163.com (X.P.); guanxfafu@126.com (X.G.)

**Keywords:** Vip3Aa, Cry1Ac, domain exchange, insecticidal activity, membrane permeability

## Abstract

*Bacillus thuringiensis* Vip3 toxins form a tetrameric structure crucial for their insecticidal activity. Each Vip3Aa monomer comprises five domains. Interaction of the first four α-helices in domain I with the target cellular membrane was proposed to be a key step before pore formation. In this study, four N-terminal α-helix-deleted truncations of Vip3Aa were produced and, it was found that they lost both liposome permeability and insecticidal activity against *Spodoptera litura*. To further probe the role of domain I in membrane permeation, the full-length domain I and the fragments of N-terminal α-helix-truncated domain I were fused to green fluorescent protein (GFP), respectively. Only the fusion carrying the full-length domain I exhibited permeability against artificial liposomes. In addition, seven Vip3Aa-Cry1Ac fusions were also constructed by combination of α-helices from Vip3Aa domains I and II with the domains II and III of Cry1Ac. Five of the seven combinations were determined to show membrane permeability in artificial liposomes. However, none of the Vip3Aa-Cry1Ac combinations exhibited insecticidal activity due to the significant reduction in proteolytic stability. These results indicated that the N-terminal helix α1 in the Vip3Aa domain I is essential for both insecticidal activity and liposome permeability and that domain I of Vip3Aa preserved a high liposome permeability independently from domains II–V.

## 1. Introduction

*Bacillus thuringiensis* (Bt) is an entomopathogen that has been used as a microbial biopesticide since 1930s [1,2]. Insecticidal Bt toxin genes have been engineered into plants to confer insect resistance for the management of insect pests. The vegetative insecticidal proteins (VIPs) from Bt have high insecticidal activities in a range of lepidopteran pests, and Bt Vip3 genes have been expressed in plants to pyramid with Bt Cry proteins to improve the protection of plants from insect damage and delay the development of resistance to Bt toxins in target pests [3,4,5].

In general, Vip3A proteins are proteolytically activated in the midgut of insect larvae, and they then bind to the midgut receptor to form pores in midgut epithelial cells [6,7,8]. Vip3A proteins contain from 786 to 803 amino acid residues with a molecular weight of around 89 kDa [9]. Previous studies have indicated that the 89 kDa Vip3A protoxin may be proteolytically cleaved by gut proteases to become active fragments before exerting insecticidal activity [10,11,12]. The formation of a Vip3Aa protein complex, composed of ~19 kDa and ~66 kDa fragments, has been identified to be required for the insecticidal activity [12,13]. As the molecular size of both the Vip3A protoxin and the activated toxin have been determined to be ~360 kDa by gel filtration, the native form of Vip3 has been believed to be a tetramer [11,13]. Recently, the molecular architectures of Vip3A and Vip3B have been revealed by Cryo-EM and X-ray crystallography to provide the tetrameric organization of both the protoxin and the trypsin-activated toxin [14,15,16]. The 3D structure of the Vip3A and Vip3B proteins also confirmed that Vip3 is composed of five distinct domains [11,14,15]. The N-terminal domain I contains four α-helices and is proposed to play a functional role in insertion of the toxin to the epithelial cell membrane [15]. Proteolytic digestion analysis also confirmed that domain I is a 19 kDa fragment of Vip3 toxins [11]. The crucial role of helix α1 in domain I has been shown for the insecticidal activity of the Vip3Aa toxin [17]. Domain II is composed of five α-helices, being the core of the tetramer that stabilizes the oligomeric structure [14,15,18]. Domain III contains three anti-parallel β-sheets, forming a β-prism fold similar to that found in Cry toxins [7,14,19]. The two C-terminal domains (domain IV and V) are highly variable in the Vip3 family and have been proposed to play roles in protecting Vip3 from digestion by proteases in the insect gut [7,15,18]. The conformational change in proteolytically activated Vip3Aa by releasing domain I α-helices from the tip of the tetrameric protoxin to form a new helical bundle at the bottom of a needle-like structure is believed to be the key step before it interacts with the membrane [14]. The stalk formed by four α-helices in domain I of trypsin-activated Vip3Ba toxins was proposed to play crucial role in the pore formation of artificial liposomes [15].

Proteolytically activated Cry1A toxin is composed of three domains [20]. Domain I, containing seven α-helices, plays a crucial role in both oligomerization and interactions with insect gut membranes. Domains II and III, rich in β-sheets, participate in receptor binding and target insect selection [21,22]. Similar to Vip3 toxins, Cry toxins were proposed to form 150- to 250-kDa oligomers after binding with receptor proteins located on the midgut epithelial membrane of target insects, although the precise mechanism of pore formation in this process remains elusive [23].

This study sheds light on the crucial role of the N-terminal helix α1 in the insecticidal activity and membrane permeation capabilities of Vip3Aa. Four N-terminal truncations of Vip3Aa were created and analyzed. Subsequently, the full-length domain I and its truncated variants were fused to green fluorescent protein (GFP) to assess their membrane permeation activity. Additionally, domains I and extensive α-helices in domain II of Vip3Aa were fused to the N-terminus of Cry1Ac domains II and III (generating Vip3Aa-Cry1Ac recombinant proteins). Although none of these recombinants exhibited insecticidal activity against *Plutella xylostella* and *Spodoptera litura* larvae, most of them displayed a high liposome permeability.

## 2. Results

### 2.1. Proteolytic Processing and Bioactivity of N-Terminus Truncated Vip3Aa Proteins

To determine the proteolytic stability of N-terminus truncated Vip3Aa proteins, four N-terminal truncations of Vip3Aa were prepared by the truncation of helix α1, helix α1 and α2, helix α1–α3, and helix α1–α4 (Figure 1a). SDS-PAGE analysis of the trypsin processed samples showed that a ~66 kDa fragment was observed in lanes containing the full-length Vip3Aa, Δα1 Vip3Aa, and Δα1–α2 Vip3Aa, respectively (Figure 1b). Numerous protein fragments, mostly ranging from 15 kDa to 66 kDa, were also observed in these three lanes showing a similar pattern, while a ~19 kDa fragment was present only in the full-length Vip3Aa samples but not in any of the four Vip3Aa truncations. Most of the protein fragments in the samples of Δα1–α3 Vip3Aa and Δα1–α4 Vip3Aa were degraded after treatment by trypsin. Only fragments at ~37 kDa, ~23 kDa, and~17 kDa were observed in the lanes containing Δα1–α3 Vip3Aa, and fragments at ~23 kDa and ~17 kDa were observed in the lanes containing Δα1–α4 Vip3Aa.

In the neonatal larval bioassays, none of the four Vip3Aa truncations exhibited toxicity in *S. litura* larvae. All the neonatal larvae that fed on a truncated Vip3Aa at 100 μg/mL became second-instar larvae after 72 h of feeding, while no neonates survived after ingesting full-length Vip3Aa (Figure 1c). The results of liposome calcein release assays indicated that none of the N-terminus truncated Vip3Aa proteins, which were processed with trypsin, had the activity to permeate liposomes, while a significant release of calcein was determined in the samples processed by the full-length Vip3Aa toxin (Figure 1d).

### 2.2. Liposome Permeability of Vip3Aa Domain I and GFP Fusion Proteins

To investigate whether the Vip3Aa domain I could independently permeabilize liposome membranes, the Vip3Aa domain I and its N-terminal truncations were fused with a GFP, respectively, creating four Vip3Aa-GFP combinations (Figure 2a). SDS-PAGE analysis indicated that the molecular weights of the four combinations were consistent with the expected sizes from 65 kDa to 49 kDa (Figure 2b). Calcein release assays were then used to assess their liposome permeation activity. Only the Trx-His-Vα1–4-GFP (containing the full-length domain I) displayed significant liposome permeabilization, in which calcein leakage began at 20 s and reached over 90% by 270 s (Figure 2c). None of the other fusion proteins carrying N-terminal truncations of the Vip3Aa domain I exhibited any detectable calcein release.

### 2.3. Bioactivity and Proteolytic Stability of Vip3Aa-Cry1Ac Recombinant Proteins

Seven recombinant proteins were generated by fusing N-terminal fragments of Vip3Aa (domain I and II) with domains II and III of Cry1Ac. These recombinants contained the Vip3Aa domain I and additional α-helices from domain II, as shown in Figure 3a. The calcein release assays revealed that five of the seven recombinants exhibited liposome permeability, while VdIα5–7-CdIIdIII and VdIdII-CdIIdIII did not (Figure 3b,c). The liposomes treated with VdI-CdIIdIII began leaking calcein within 400 s, reaching 100% leakage by 680 s. Interestingly, the calcein release curve for the VdI-CdIIdIII-treated liposomes closely resembled that of the liposomes treated with trypsin-activated Vip3Aa. While VdI-CdIIdIII and trypsin-activated Vip3Aa exhibited similar potencies, the other permeable fusion proteins (VdIα5-CdIIdIII, VdIα5–6-CdIIdIII, VdIα5–8-CdIIdIII, and VdIα5–9-CdIIdIII) displayed lower calcein release rates. Despite their membrane permeation activity, none of the seven Vip3Aa-Cry1Ac fusions exhibited insecticidal activity against *S. litura* or *P. xylostella* larvae. Feeding larvae with these fusion proteins resulted in nearly no mortality, while Vip3Aa and Cry1Ac individually caused 100% and 75% mortality in *S. litura* (Figure 4a) and 97% and 100% mortality in *P. xylostella* (Figure 4b), respectively.

To determine proteolytic stability of these Vip3Aa-Cry1Ac recombinants, each recombinant protein was incubated with trypsin in vitro and analyzed by SDS-PAGE. Although part of the recombinant proteins was degraded into fragments at a molecular weight between 25 kDa and 70 kDa, the majority of them remained integrated at a molecular weight from 80.8 kDa to 95.1 kDa after the treatment without trypsin (Figure 5a). Processing the Vip3Aa protoxin with trypsin resulted in a ~66 kDa fragment, a ~20 kDa fragment, and multiple protein fragments between 25 kDa and 40 kDa (Figure 5b). A significant band at ~60 kDa was observed in the lane containing gut proteases processed Cry1Ac protein. In comparison, all the Vip3Aa-Cry1Ac recombinants were degraded, and no protein bands could be observed in the SDS-PAGE analysis.

## 3. Discussion

Cryo-EM studies have revealed the tetrameric structure of Vip3Aa protoxin and trypsin-activated toxin, with each monomer being composed of five domains [14]. Domain I, containing four α-helices, plays a crucial role in both oligomerization and insecticidal activity [17]. This domain undergoes a conformational change during proteolytic activation, where helices α1 to α3 unfold and form a new N-terminal coiled-coil interacting with membranes [15]. To investigate the functional roles of these helices, we generated truncated Vip3Aa proteins lacking α1, α1–α2, α1–α3, or α1–α4. The trypsin treatment revealed the significant impact of these α-helices on proteolytic activation. Helix α1–α3 and α1–α4 deletions led to complete degradation, highlighting their essential role in maintaining protein stability during proteolysis (Figure 1b). The deletion of helix α1 or α1–α2 resulted in fragmentation, similar to the protoxin, but it lacked the characteristic ~19 kDa fragment, which has been known as the fragment of released domain I and plays significant role in insecticidal activity [7,11]. Functionally, truncating either helix α1 or a larger N-terminal segment abolished both the liposome permeation and insecticidal activity against *S. litura* (Figure 1c,d). These observations agree with previous findings on N-terminal deletions abolishing insecticidal activity [24,25] and the crucial role of helix α1 in Vip3Aa’s structure and function [17]. Structural analyses suggest that domain I–III maintains the spontaneous conformation of both the protoxin and activated toxin [11]. In this study, deleting the first three α-helices (α1–α3) likely destabilized the protein, leading to complete degradation by trypsin or midgut proteases. The deletion of the first one or two α-helices, however, preserved the proteolytic stability but prevented r processing into active fragments. These findings indicate that the first three N-terminal α-helices of Vip3Aa domain I are not only functional in pore formation but also crucial for maintaining structural stability during proteolytic processing.

Previous studies have estimated the crucial role of Vip3Aa domain I in membrane permeability [11,17]. We further verified this by fusing domain I with GFP, demonstrating its independent activity in membrane perturbation. Only the liposomes treated with the full-length domain I fusion showed significant calcein release (Figure 2c). This confirms that domain I can independently perforate artificial liposome membranes, while the lack of activity in N-terminal truncations underscores the critical role of helix α1 in membrane permeability. Despite the Trx-His tag promoting solubility and proper folding [26], the Vip3Aa-GFP fusions still aggregated and formed inclusion bodies. However, the high activity of renatured Trx-His-Vα1–4-GFP in permeating liposomes highlights the highly hydrophobic nature of Vip3Aa domain I.

The striking structural homology between the Vip3 and 3D-Cry toxins suggests similar functional roles for their respective domains [15]. Similar to Vip3A domain I, Cry1Ac domain I is responsible for pore formation [27,28]. Additionally, domains II and III of Cry1Ac are involved in receptor binding, target selection, and structural maintenance, analogous to the proposed roles of domains III–V in Vip3A [7,29,30,31]. According to the individual insecticidal activity of Vip3Aa and Cry1Ac against *S. litura* and *P. xylostella* larvae [12,32,33,34], we fused domain I and additional α-helices in domain II of Vip3Aa with domains II and III of Cry1Ac to generate multiple chimeric proteins (Figure 3a). The calcein release assays showed that apart from VdIα5–7-CdIIdIII and VdIdII-CdIIdIII, the other recombinant proteins exhibited membrane permeability against artificial liposomes (Figure 3b,c). We hypothesize that the highly hydrophobic helix α1 of Vip3Aa might be folded and become unexposed due to structural changes in VdIα5–7-CdIIdIII and VdIdII-CdIIdIII, leading to the loss of liposome permeation activity. The fusions of Vip3Aa domain I with either GFP or Cry1Ac domains were found to be largely insoluble in native conditions. Urea denaturation and the subsequent renaturation may not lead to the formation of the tetrameric complex as the full-length Vip3Aa toxin. The activity of these fusions to rupture artificial liposome membranes indicated that the monomeric form of Vip3Aa domain I, the core component of the coiled-coil structure in the Vip3Aa tetramers, possesses sufficient membrane-destabilizing activity even in the absence of pore formation. The α-helical architecture of Vip3Aa domain I might facilitate direct interaction with the phospholipid bilayer, similar to the observed mechanism of colicins interacting with phospholipid vesicles [35,36]. Previous studies have investigated the permeability of both Cry1A and Vip3A toxins against artificial liposomes [37,38,39]. The highly hydrophobic α-helices in domain I have been associated to the pore-forming activity of Cry1A and Vip3A toxins [15,27,40]. Cry1A’s activity was shown to be enhanced when liposomes contained soluble BBMV (brush border membrane vesicle) proteins, representing potential Cry1Ac toxin receptors [38]. In this study, we focused solely on assessing the membrane disruption activity of fusion proteins carrying Vip3Aa domain I against artificial membranes. However, the epithelial membrane in the midgut of lepidopteran insects is undoubtedly more complex than a simple liposome model. Therefore, future research is necessary to determine the permeability of these recombinant proteins in eukaryotic cells expressing specific receptors. 

Chimeric proteins composed of the full-length Vip3Aa and toxic core of Cry1Ac resulted in a 150 kDa chimera showing increased toxicity against *Ephestia kuehniella* [41]. Fusing Vip3Aa7 to the N-terminus of Cry9Ca also improved insecticidal activity against *P. xylostella* [33]. In this study, although the liposome permeability of these Vip3Aa-Cry1Ac recombinant proteins were determined, none of them displayed insecticidal activity against either *S. litura* or *P. xylostella* larvae (Figure 4a,b). The proteolytic processing results showed a significant degradation of most of the recombinant proteins after treatment by trypsin (Figure 5b). In addition to VdI-CdIIdIII and the other Vip3Aa-Cry1Ac recombinant proteins containing domain I and extension α-helices in domain II of Vip3Aa, the trypsin cleavage site between domain I and domain II was consequently involved. Exposure of this trypsin cleavage site in these fusion proteins may be responsible for the degradations. However, the recombinant protein of VdI-CdIIdIII without this trypsin cleavage site was also fully degraded. These results indicated that fusing the N-terminal domains of Vip3Aa with the C-terminal domains of Cry1Ac may have led to the loss of protection of the active fragments under the proteolytic processing. This drastic reduction in proteolytic stability could be a potential reason for the loss of insecticidal activity.

## 4. Conclusions

In conclusion, this study highlights the crucial roles of N-terminal α-helices in Vip3Aa domain I for their proteolytic stability and membrane permeation. Domain I is a highly hydrophobic fragment. Renatured domain I fragments could show activity in membrane permeating against liposomes without the help of other domains in Vip3Aa. The combination of α-helices in domains I and II with domains II and III of Cry1Ac eliminated insecticidal activity, while the liposome permeability of most of the recombinant proteins remained.

## 5. Materials and Methods

### 5.1. Construction of Expression Vectors

The full-length *vip3Aa* gene (NCBI accession no. AF500478.2) derived from the Bt WB5 strain producing Vip3Aa12 protein was exercised from plasmid pGEX-Vip3Aa and cloned into the vector pET32a at the downstream of a 3C PreScission protease cleavage site [42]. To construct pET32a expression vectors expressing truncated Vip3Aa, fragments of *vip3Aa* with different truncations in the domain-I-coding region were amplified from pGEX-Vip3Aa by PCR using the iProof^TM^ High-Fidelity Master Mix DNA polymerase (Bio-RAD, Hercules, CA, USA) with the primer sets listed in Table 1. The PCR-amplified fragments were cloned to pET32a at the downstream of a 3C PreScission protease cleavage site, as described above. Four expression vectors were constructed to express Vip3Aa with a truncation of the α1 to α4 helices in domain I of Vip3Aa (Figure 1a).

Four fragments of Vip3Aa domain I, including the full-length domain and three N-terminal truncations, were cloned, and fused to the 5’ end of the *gfp* gene using overlap PCR [43]. Specific primer sets listed in Table 1 facilitated the construction of these fusions. The resulting PCR products containing the truncated Vip3Aa domains and *gfp* were then inserted into the pET32a expression vector at the downstream of a Trx-His tag. Four expression vectors were constructed to prepare fusion proteins: Trx-His-Vα1–4-GFP (full-length domain I), Trx-His-VΔα1-GFP (α1-truncated), Trx-His-VΔα1–α2-GFP (α1 and α2-truncated), and Trx-His-VΔα1–α3-GFP (α1 to α3-truncated). Schematic representations of these fusion proteins are presented in Figure 2a.

To generate Vip3Aa-Cry1Ac combinations, nucleotide fragments encompassing Vip3Aa domains I and II (VdIdII) and its C-terminal truncations were amplified using primers listed in Table 2. Plasmid DNA from Bt HD73 strain, expressing the Cry1Ac1 protein (NCBI Accession No. AAA22331.1) was extracted as described previously [44]. Fragments containing domains II and III of Cry1Ac (CdIIdIII) were also amplified using specific primers (Table 2). Overlap PCR, as described above, facilitated the ligation of Vip3Aa and Cry1Ac fragments. The resulting PCR products were then inserted into pET32a at the downstream of the Trx-His tag, generating seven recombinant proteins: VdI-CdIIdIII (Vip3Aa domain I, without the K198 trypsin cleavage site, fused with Cry1Ac domains II and III), VdIα5-CdIIdIII (α5-extended at the C-terminus of Vip3Aa domain I), VdIα5–6-CdIIdIII (α5–6 extended at the C-terminus of Vip3Aa domain I), VdIα5–7-CdIIdIII (α5–7 extended at the C-terminus of Vip3Aa domain I), VdIα5–8-CdIIdIII (α5–8 extended at the C-terminus of Vip3Aa domain I), VdIα5–9-CdIIdIII (α5–9 extended at the C-terminus of Vip3Aa domain I), and VdIdII-CdIIdIII (Vip3Aa domains I and II fused with Cry1Ac domains II and III). The expression vectors for these Vip3Aa-Cry1Ac combinations are schematically illustrated in Figure 3a.

### 5.2. Expression and Purification of Recombinant Proteins

To prepare Vip3Aa proteins, a 250 μL overnight culture of *Escherichia coli* BL21 (DE) cells carrying a plasmid of pET32a-Vip3Aa was inoculated to 400 mL of LB in a 2 L flask. The cultures were incubated at 37 °C and 150 rpm in a shaking incubator until the OD_600_ reached 0.5. Protein expression was then induced by adding 0.8 mM IPTG (Isopropyl-D-thiogalactoside), followed by incubation at 16 °C for 24 h. Cells were harvested by centrifugation at 14,000× *g* for 1 min and resuspended in Tris–NaCl buffer (50 mM Tris, 0.5 M NaCl, pH 8.6) for washing. The cell suspension, supplemented with 1 mM PMSF (phenylmethanesulfonyl fluoride), was disrupted by sonication using a VC-50 sonicator (Sonics & Materials Inc., Newtown, CT, USA). Cell debris was pelleted by centrifugation at 21,000× *g* for 10 min. The supernatant containing soluble Trx-His-3C-Vip3Aa fusion protein was purified using a Ni-Sepharose 6 Fast Flow column (Cytiva, Shanghai, China) according to the manufacturer’s instructions. The fusion protein was eluted with 500 mM imidazole in 50 mM Tris–NaCl buffer (pH 8.6). The recombinant proteins after dialysis were collected and reloaded to the Ni-Sepharose column to remove cleaved Trx-His tag and residual fusion proteins. The GST-tagged 3C PreScission protease used for cleavage of the fusion proteins was removed from the recovered recombinant proteins using a GSTrap column (Cytiva, Shanghai, China).

To prepare the Vip3Aa-GFP and Vip3Aa-Cry1Ac fusion proteins, *E. coli* BL21 (DE3) cells carrying the corresponding plasmids were cultured and induced with 0.8 mM IPTG as described above. After harvesting, the cells were washed and resuspended in Tris–NaCl buffer (pH 8.6) containing 1 mM PMSF. Cell suspensions were sonicated, and the inclusion bodies containing insoluble target proteins were collected by centrifugation. These inclusion bodies were dissolved in 6 M urea solution (50 mM Tris, pH 7.5), and the resulting solution was loaded onto a Ni-Sepharose 6 Fast Flow column for purification of the Trx-His-tagged recombinant proteins. Elution was achieved with 3.6 M urea solution containing 500 mM imidazole in 50 mM Tris–NaCl buffer (pH 8.6). Renaturation of the target proteins was carried out through a stepwise urea gradient dialysis using buffers containing 3 M, 2 M, 0.5 M, and 0 M urea in Tris–NaCl buffer (pH 8.6). After dialysis, the supernatant containing soluble recombinant proteins was collected by centrifugation at 12,000× *g* for 2 min at 4 °C and stored at −20 °C for further use.

### 5.3. Insect Rearing and Bioassays

An inbred lab colony of *S. litura* maintained in the laboratory for over 4 years (~40 generations) [12] was used in this study. The *S. litura* colony was reared on a soybean-based artificial diet at 27 °C with 50% humidity and a photoperiod 16 h of light and 8 h of darkness. The larvae of Bt-susceptible *P. xylostella* was kindly provided by the Institute of Applied Ecology (IAE), Fujian Agriculture and Forestry University [45].

For bioassays using trypsin-processed proteins, assayed proteins were processed by bovine pancreas trypsin (Sigma, St. Louis, MO, USA) at a mass ratio of 1.2:1 (trypsin/Vip3Aa, *w*/*w*) at 37 °C for 6 h. Bioassays were conducted using a diet overlay method [46]. Briefly, assayed proteins were diluted in distilled water to 100 μg/mL, respectively. A 200 μL aliquot of diluted proteins was overlaid on the surface (~7 cm^2^) of the diet in each cup (30 mL plastic cup containing ~5 mL diet). Ten neonates of *S. litura* or 2nd-instar larvae of *P. xylostella* were placed into each cup, and each concentration of Vip3Aa included 5 replications. Cups containing diet overlaid by distilled water were used as negative controls. Diet cups containing assayed larvae were covered with lids and kept in a rearing room at 27 °C and at 50% humidity and with a photoperiod of 16:8 (light–dark). The mortality of the assayed *S. litura* and *P. xylostella* was recorded in 72 h and 48 h, respectively.

### 5.4. In Vitro Proteolytic Processing of Recombinant Proteins

In vitro proteolytic processing was conducted as described previously [12,47]. To investigate the proteolytic stability of Vip3Aa truncations and Vip3Aa-Cry1Ac recombinant proteins, 17.5 μg of Vip3Aa truncations was incubated with bovine pancreas trypsin at ratios of 1.2:1 (trypsin/Vip3Aa, *w*/*w*) in 100 μL of 150 mM NaCl in 50 mM Tris–HCl buffer (pH 8.6) for 1 h at 37 ℃. Target proteins incubated in buffer only were used as controls. Proteolytically processed proteins were separated by electrophoretic analysis on 15% SDS-PAGE gels.

### 5.5. Liposome Fluorescence Permeability Assay of Vip3Aa Complex Activities

The activity of the target proteins to permeate the lipid membrane was determined using liposomes for a calcein release assay [37,38,48]. To prepare liposomes, 1,2-dioleoyl-sn-glycero-3-phosphocholine (DOPC) and 1,2-dioleoyl-sn-glycero-3-phosphoethanolamine (DOPE) were mixed at a ratio of 1:1 (*w*/*w*), and the lipid mixture was dissolved in chloroform at a final concentration of 10 μM/mL. The solvent (chloroform) in the lipid solution was evaporated with a flow of nitrogen gas and then under vacuum in a lyophilizer for 2.5 h. Approximately 15.2 mg of dried lipids was suspended in 200 μL of 30 mM calcein in 50 mM Tris–HCl at pH 8.0. After 5 cycles of freezing and thawing, the solution was passed through a polycarbonate membrane (0.1 μM pore size) for 35 passes using a two-syringe extruder (Avanti Polar Lipid, Alabaster, AL, USA). The lipid vesicle solution was loaded to a HiTrap desalting column with Sephadex G-25 resin (Cytiva) to remove free calcein, and the liposome preparation was recovered and stored until use.

To examine the membrane permeation activity of Vip3Aa truncations, the calcein-encapsulated liposome solution was carefully pipetted to a 1.0 cm light-path quartz cuvette and then mixed with MnCl_2_ at a final concentration of 10 μM. Twenty microliters of trypsin-processed Vip3Aa truncations at a concentration of 0.5 μM were added to the cuvette containing 250 μL of calcein-encapsulated liposome solution. After an incubation of the mixture for 300 s for to allow the liposomes to stabilize, the release of calcein from the liposomes was monitored as decreasing in fluorescence for 10 min using a fluorescence spectrophotometer (F-4600, HITACHI, Tokyo, Japan) at excitation and emission wavelengths of 485 and 520 nm, respectively, with a slit width of 5 nm. Triton X-100 (0.1%, *v*/*v*) was added to the cuvette to solubilize liposomes at the end of the assay to completely release the calcein. THe liposomal permeability of Vip3Aa-GFP fusions or Vip3Aa-Cry1Ac recombinant proteins without being processed with trypsin was assayed through the same procedure described above. Trx-His tag was used in the assays as a negative control. At least three replications were included for each assay. The percentage of calcein released from the vesicles (I%) was calculated by the following formula: I% = 100 × (I_t_ − I_0_)/(I_max_ − I_0_). I_0_, I_t_, and I_max_ represent the intensity of the fluorescence of the original calcein-encapsulated lipid solution without assayed protein samples, the calcein-encapsulated lipid solution with an assayed protein sample, and the sample with 0.1% Triton X-100 added, respectively.

## Figures and Tables

**Figure 1 toxins-16-00088-f001:**
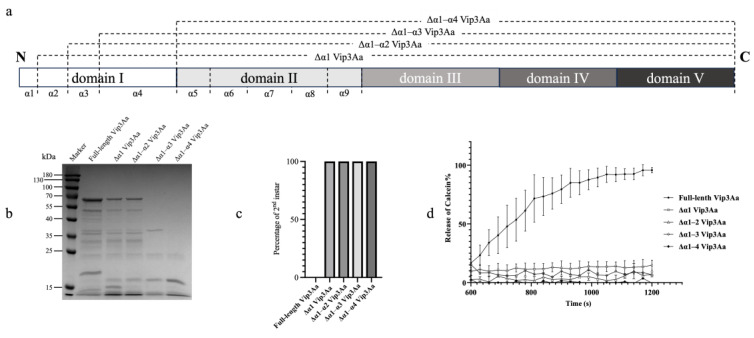
Proteolytic stability and activity assays of four Vip3Aa truncations. Panel (**a**): Schematic presentation of four N-terminal truncations of Vip3Aa proteins. Four N-terminal truncations of Vip3Aa were prepared by truncation of helix α1, helix α1 and α2, helix α1–α3, and helix α1–α4. Panel (**b**): SDS-PAGE analysis of four Vip3Aa truncations after tryptic processing at a ratio of 1.2:1 (trypsin–Vip3Aa, *w*/*w*) at 37 °C for 1 h. Panel (**c**): Neonates of *S. Litura* were fed with 100 μg/mL of N-terminal truncations of Vip3Aa and the full-length Vip3Aa, respectively, by surface overlay bioassays. Numbers of larvae developed to the second instar were recorded after 72 h. Panel (**d**): Calcein release assays by treating liposomes with 16 nM of four N-terminal truncations of Vip3Aa after tryptic processing. Fluorescence signals of released calcein were recorded every 30 s. The error bars represent the standard error of mean of the measurements from triplicate experiments.

**Figure 2 toxins-16-00088-f002:**
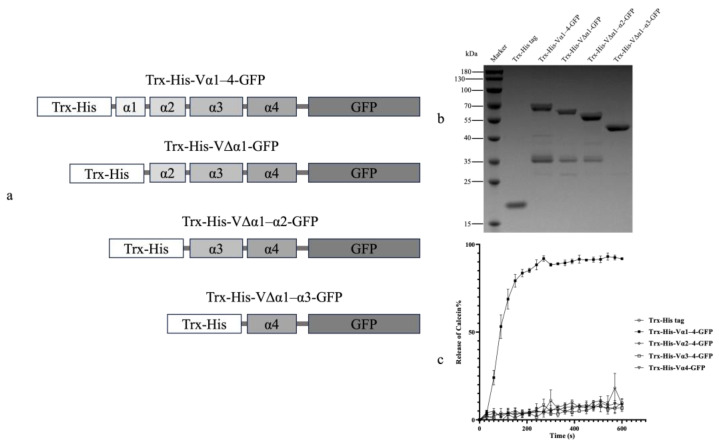
Calcein release assays by treating liposomes with Vip3Aa-GFP fusion proteins. Domain I of Vip3Aa and its N-terminal truncations were fused with a GFP and a Trx-His tag to produce four fusion proteins. Panel (**a**): schematic presentation of four Vip3Aa-GFP recombinant proteins. Panel (**b**): each fusion protein was purified and loaded on an SDS-PAGE gel to determine their molecular weight. Panel (**c**): four Vip3Aa-GFP fusion proteins and Trx-His tag protein were added to the liposome solution (DOPC–DOPE, 1:1) at a final concentration of 10 μM/mL, respectively. Fluorescence signals of released calcein were recorded every 30 s. The error bars represent the standard error of mean of the measurements from triplicate experiments.

**Figure 3 toxins-16-00088-f003:**
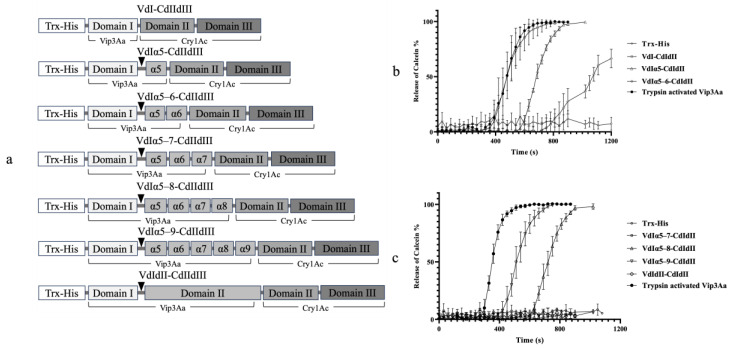
Calcein release assays by treating liposomes with Vip3Aa-Cry1Ac fusion proteins. Domain I and domain II extensions of Vip3Aa were fused with a Trx-His tag and domains II and III of Cry1Ac to produce seven Vip3Aa-Cry1Ac recombinant proteins. The trypsin cleavage site between domain I and II of Vip3Aa is presented by a triangle. Liposome (DOPC–DOPE, 1:1) permeability of seven Vip3Aa-Cry1Ac fusion proteins was determined through calcein release assays. The final concentration of each assayed protein was 10 μM/mL. Fluorescence signals of released calcein were recorded every 30 s. The error bars represent the standard error of mean of the measurements from triplicate experiments. Trx-His tag and trypsin-activated Vip3Aa were used as the negative control and the positive control, respectively. Panel (**a**): schematic presentation of seven Vip3Aa-Cry1Ac recombinant proteins. Panel (**b**): calcein release rate of recombinant proteins, including VdI-CdIIdIII, VdIα5-CdIIdIII, and VdIα5–6-CdIIdIII. Panel (**c**): calcein release rate of recombinant proteins, including VdIα5–7-CdIIdIII, VdIα5–8-CdIIdIII, VdIα5–9-CdIIdIII, and VdIdII-CdIIdIII.

**Figure 4 toxins-16-00088-f004:**
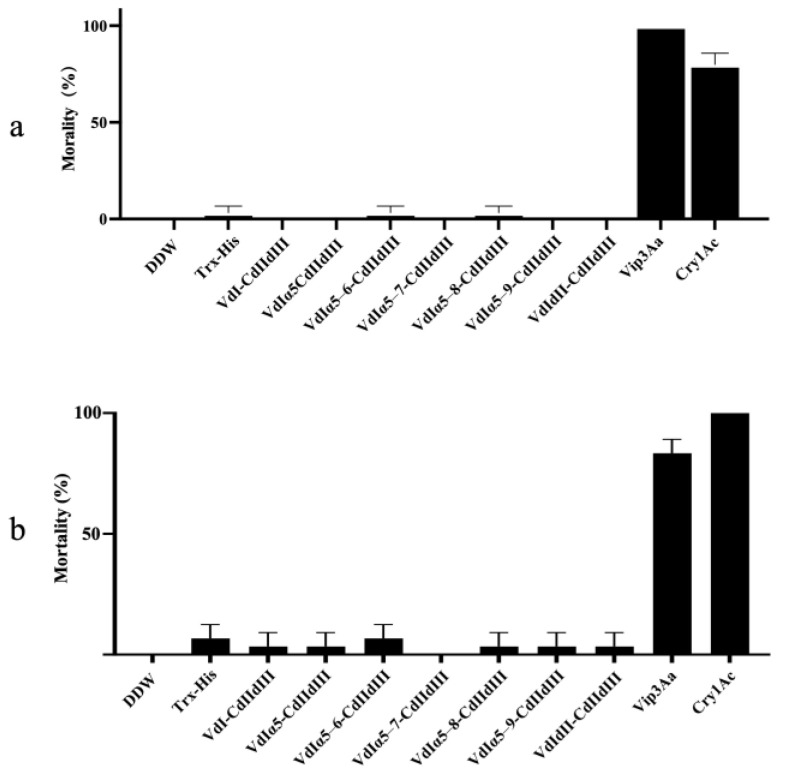
Mortality of *Spodoptera litura* and *Plutella xylostella* larvae by feeding with seven Vip3Aa-Cry1Ac recombinant proteins. Panel (**a**): Neonates of *S. litura* were fed with 100 μg/mL of seven Vip3Aa-Cry1Ac recombinant proteins, respectively, by surface overlay bioassays. Mortality was recorded after 72 h. Panel (**b**): The second-instar larvae of *P. xylostella* were fed with 100 μg/mL of seven Vip3Aa-Cry1Ac recombinant proteins, respectively, by surface overlay bioassays. Mortality was recorded after 48 h. Mortality of larvae fed with the full-length Vip3Aa and Cry1Ac toxin, respectively, was recorded as the positive controls. The error bars represent the standard error of mean of the mortality from triplicate-assayed groups. DDW: double-distilled water.

**Figure 5 toxins-16-00088-f005:**
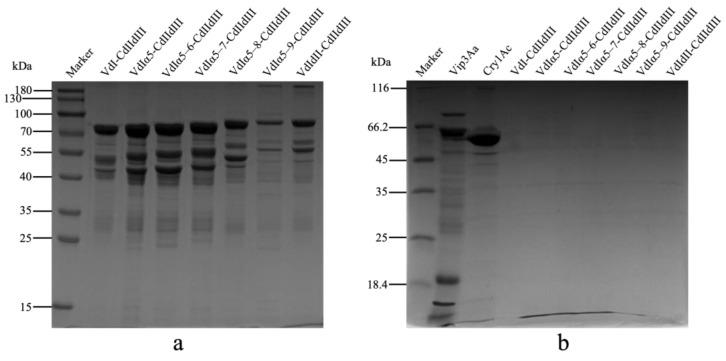
Proteolytic stability of seven Vip3Aa-Cry1Ac recombinant proteins. Panel (**a**): Seven Vip3Aa-Cry1Ac recombinant proteins were incubated at 30 °C for 6 h without any proteases and loaded to a 15% SDS-PAGE gel, respectively. Panel (**b**): seven Vip3Aa-Cry1Ac recombinant proteins, the full-length Vip3Aa, and Cry1Ac protoxins were incubated with bovine pancreas trypsin at ratios of 1.2:1 (trypsin–Vip3Aa, *w*/*w*) for 1 h at 37 °C, respectively, and loaded to a 15% SDS-PAGE gel to separate the digested fragments.

**Table 1 toxins-16-00088-t001:** Primers for the cloning of vectors expressing Vip3Aa N-terminal truncations and Vip3Aa-GFP fusions.

Primers	Sequences (5′ to 3′)
Vip3Aa-F	acaaggccatggctgatatcGGATCC*CTGGAGGTGCTGTTCCAGGGGCCC*ATGAACAAGAATAATACTAAATTA
Δα1 Vip3Aa-F	acaaggccatggctgatatcGGATCC*CTGGAAGTTCTGTTCCAGGGGCCC*ACGGATACAGGTGGTGATCTAA
Δα1–α2 Vip3Aa-F	acaaggccatggctgatatcGGATCC*CTGGAAGTTCTGTTCCAGGGGCCC*GGAAACTTAAATACAGAATTAT
Δα1–α3 Vip3Aa-F	acaaggccatggctgatatcGGATCC*CTGGAAGTTCTGTTCCAGGGGCCC*AAGTTGGATATTATTAATGTAA
Δα4 Vip3Aa-F	acaaggccatggctgatatcGGATCC*CTGGAAGTTCTGTTCCAGGGGCCC*GAAACTAGTTCAAAAGTA
Vip3Aa-R	agtggtggtggtggtggtg CTCGAGTTACTTAATAGAGACATCGT
Vip3A-DI-GFP-R	*agtgaaaagttcttctcctttactcat*AGCAAAAGTTAATTCCTCAAATT
GFP-F	TTGAGGAATTAACTTTTGCT*atgagtaaaggagaagaacttttcact*
GFP-R	gttagcagccggatctcagtggtggtggtggtggtgCTCGAG*ttatttgtatagttcatccatgccatgt*

Sequences overlap to pET32a are lowercased; cleavage sites of *Bam*H I, *Xho* I are underlined; cleavage site of 3C PreScission protease is in italics; sequences overlap to *gfp* are lowercased and in italics.

**Table 2 toxins-16-00088-t002:** Primers for the cloning of vectors expressing Vip3Aa-Cry1Ac chimeric proteins.

Primers	Sequences (5′ to 3′)
Vip3Aa-F	acaaggccatggctgatatcGGATCC*CTGGAGGTGCTGTTCCAGGGGCCC*ATGAACAAGAATAATACTAAATTA
VdI-Cry-R	aattgggaaactgttcgaattggTTCTGTAGCAAAAGTTAAT
VdI-Cry-F	TTCTTGATGAGTTAACTGAGccaattcgaacagtttcccaatt
VdIα5-Cry-R	aattgggaaactgttcgaattggTACACTTTTCGCTAGTTCAGTT
VdIα5-Cry-F	TTAACTGAACTAGCGAAAAGTGTAccaattcgaacagtttcccaatt
VdIα5–6-Cry-R	aattgggaaactgttcgaattggCATTACATCGTGGAATGTATTAA
VdIα5–6-Cry-F	TTAATACATTCCACGATGTAATGccaattcgaacagtttcccaatt
VdIα5–7-Cry-R	aattgggaaactgttcgaattggATTTTCTTTAGTAATTAATTCC
VdIα5–7-Cry-F	GGAATTAATTACTAAAGAAAATccaattcgaacagtttcccaatt
VdIα5–8-Cry-R	aattgggaaactgttcgaattggTAATAATTTTCGGCATGTTGTTAA
VdIα5–8-Cry-F	TTAACAACATGCCGAAAATTATTAccaattcgaacagtttcccaatt
VdIα5–9-Cry-R	aattgggaaactgttcgaattggGTTTACTCTAAATTCCTCTTTTTC
VdIα5–9-Cry-F	GAAAAAGAGGAATTTAGAGTAAACccaattcgaacagtttcccaatt
VdIdII-Cry-R	aattgggaaactgttcgaattggAGAAAGTGTAGGGAGGATGTTTACTC
VdIdII-Cry-F	GAGTAAACATCCTCCCTACACTTTCTccaattcgaacagtttcccaatt
Cry1Ac-R	gtggtggtggtggtggtgctcgagttatgcagtaactggaataaattcaaatc

Sequences overlapping pET32a are lower-case; cleavage sites of *Bam*H I, *Xho* I are underlined; cleavage site of 3C PreScission protease is in italics; sequence overlapping *cry*1*Ac* are lower-case and underlined.

## Data Availability

The data presented in this study are available on request from the corresponding author Li Sha (shal@fafu.edu.cn).

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
