# Peer review of "N-Terminal α-Helices in Domain I of Bacillus thuringiensis Vip3Aa Play Crucial Roles in Disruption of Liposomal Membrane"

_toxins, 2024, doi:10.3390/toxins16020088_

Round 1

Reviewer 1 Report

Comments and Suggestions for Authors

This manuscript describes the production and testing of various deletions and fusions involving the N-terminal helical region of the Vip3Aa toxin in order to determine its role in pore formation and insect toxicity.  These constructs included Vip3Aa-Cry1Ac fusions to investigate the behaviour of fusions between these toxins that share some structural similarities.  It would have been interesting to test by size exclusion chromatography, whether these fusions form tetramers or not (particularly those with intact domain II regions. 

The work is an interesting contribution to the field.  I have only minor suggestions and requests for clarifications.

General: The authors should make clear which variant of Vip3Aa they are using throughout this work.  Similarly, they should specify which Cry1Ac variant is used in the fusions.

Line 56: “syringe-like” may suggest some sort of contractile injection mechanism (as seen with toxin complex proteins).  I would suggest needle-like

Line 64: I do not understand the use of “proper” here

Sections 2.2 and 2.3: Although the constructs are described in the methods and supplementary figures, the format of Toxins papers with methods at the end means that at this point in the text, it is hard to know exactly what each of the constructs contains.  I think it might be useful to bring the figures from supplementary into the main text so that it is clear what constructs are being analyzed without reference to other material.  It might also be useful to include annotation of a5-8 in at least one figure (current figure S1?) so that readers can understand eg VdIa5-8-CdIIdIII.

The authors should also clarify whether the GFP and Cry fusions were trypsinized before use in the calcein release assays.

Line 144: record -> recorded

Line 159: proteins ->protein

Tables 1 & 2: would be better placed in the methods section

Lin2 228 says that VdIa5-8-CdIIdIII did not form pores in liposomes but line 126 says VdIa5-7-CdIIdIII (as suggested in Figure 3).  Which is correct?

Line 250: delete “of”

Line 260: “I” -> In

Line 362: gly-cero -> glycero?

Comments on the Quality of English Language

Minor grammatical issues only.  Does not prevent clear understanding

Reviewer 2 Report

Comments and Suggestions for Authors

This manuscript shows a study of the functional role of Vip3Aa domain I by deleting several of its alpha helices and making fusion proteins with GFP and Cry1Ac. The results show the importance of alpha helix 1 in pore formation and insecticidal activity. The integrity of most of domain I is also important for protease stability.

Main concern:

The fusion proteins, with either GFP or Cry1Ac, cannot form the coiled-coil structure that domain I forms in the Vip3A protein in solution. However, the results show that the unstructured domain I can be functional realising calcein in the liposome assay (Fig. 2b and Fig. 3). This is an evidence that just the alpha-helices of domain I are sufficient to destabilize the liposomal membrane. It is important that the authors discuss that this is the mechanism of other pore-forming toxins that do not form real pores but destabilize the membrane, such as colicine. It would also make the results more clear if the authors stress, in the Discussion section, that these results are obtained with the domain I in monomeric form and that, in vivo, Vip3 domain I is found in the tetrameric coiled-coil form.

Other comments:

Line 44: Reference 14 did not provide a tetrameric structure, but just dimeric. It has to be replaced by the publication of Zheng, M., Evdokimov, A. G., Moshiri, F., Lowder, C., Haas, J., 2020. Crystal structure of a Vip3B family insecticidal protein reveals a new fold and a unique tetrameric assembly. Prot. Sci., 29, 824829.

Line 45: The appropriate reference is not 7 but 11.

Fig 2a: What is the band in lane 3?

Fig. 2b: The legend on the right side is incorrect, since according to it, the construct lacking alpha1 is the only one producing calcein release. This does not agree with the text.

Fig. 4: Define, in the text of the legend, the meaning of DDW (double distilled water?).

Fig. 5: The legend of 5b refers to larvae gut extract, whereas the text of the manuscript (lines 158-164) is talking about treatment with trypsin. What is correct? Line 250 also says trypsin.

Tables 1 and 2 appear before being cited in the text.

Line 227: It says Complemental figure when it has to say Supplemental figure.

Line 278: Figure S1 cannot be cited after figs. S2 and S3. Move it up.

Line 342: Change distil for distilled.

Lines 350-358: This section does not describe the incubation with larvae midgut protease extract.

Lines 375-376: How much volume of the proteins was added to the 250 ul of liposome solution? Did it significantly change the concentration of liposomes and thus, the fluorescence?

Round 2

Reviewer 2 Report

Comments and Suggestions for Authors

I'm happy with the changes introduced.

Supplemental figures 1, 2 and 3 are now integrated into the body of the manuscript